# Gradient-based Gradual Pruning for Language-Specific Multilingual Neural Machine Translation

**Dan He, Minh-Quang PHAM, Thanh-Le Ha, Marco Turchi**

Zoom Video Communications

{dan.he,minhquang.pham,thanhle.ha,marco.turchi}@zoom.us

## Abstract

Multilingual neural machine translation (MNMT) offers the convenience of translating between multiple languages with a single model. However, MNMT often suffers from performance degradation in high-resource languages compared to bilingual counterparts. This degradation is commonly attributed to parameter interference, which occurs when parameters are fully shared across all language pairs. In this work, to tackle this issue we propose a gradient-based gradual pruning technique for MNMT. Our approach aims to identify an optimal sub-network for each language pair within the multilingual model by leveraging gradient-based information as pruning criterion and gradually increasing the pruning ratio as schedule. Our approach allows for partial parameter sharing across language pairs to alleviate interference, and each pair preserves its unique parameters to capture language-specific information. Comprehensive experiments on IWSLT and WMT datasets show that our approach yields a notable performance gain on both datasets.

## 1 Introduction

In recent years, neural machine translation (NMT) based on the transformer architecture has achieved great success and become the dominant paradigm for machine translation (Sutskever et al., 2014; Vaswani et al., 2017). Multilingual neural machine translation (MNMT), which learns a unified model to translate between multiple languages, has attracted growing attention in the NMT area (Ha et al., 2016; Johnson et al., 2017). The reasons are: 1) From a practical perspective, it significantly reduces the training and inference cost and simplifies deployment in production; 2) Utilizing data from multiple language pairs simultaneously can potentially help improve the translation quality of low-resource or even zero-resource language pairs by transferring knowledge across languages.

Despite these benefits, MNMT remains challenging as its performance degrades compared to bilingual counterparts in high-resource languages (Arivazhagan et al., 2019). Previous studies (Wang et al., 2020; Shaham et al., 2023) attribute this degradation to parameter interference: each language is unique and therefore has distinct requirements from the model parameters to capture language-specific information, and fully sharing parameters causes negative interactions across language pairs as languages compete for model capacity. Although naively increasing model size may alleviate parameter interference, large models often suffer from parameter inefficiency and overfitting (Zhang et al., 2020; Arivazhagan et al., 2019).

To address the parameter interference issue more effectively, researchers have explored various methods to allocate language-specific parameters to capture unique linguistic information for each language pair. One line of work focused on introducing extra parameters to the model. For instance, adapter-based approaches (Bapna and Firat, 2019; Philip et al., 2020; Zhu et al., 2021; Baziotis et al., 2022) inject lightweight language-specific modules into the shared model. Another well-known method (Dong et al., 2015) utilizes a shared encoder but a different decoder for each target language. Despite being effective, these methods can become parameter-inefficient as the number of languages in the multilingual model grows.

Another line of work focused on extracting a separate sub-network within the multilingual model for each language pair. In these approaches, each sub-network preserves exclusive parameters to capture the language-specific features and has some parameters shared with other languages (Xie et al., 2021; Lin et al., 2021; Wang and Zhang, 2022; Pham et al., 2022). A recent work (Lin et al., 2021) shows promising results by first training a multilingual model that covers all language pairs, then fine-tuning the trained model on each pair, and in

the end, extracting the sub-networks via magnitude pruning in one step. Although this paradigm is intuitive and straightforward, it can be suboptimal due to the following limitations: (1) Magnitude pruning is demonstrated to be ineffective in a transfer learning context (Sanh et al., 2020), potentially affecting NMT systems in the multilingual scenario. (2) Pruning in one single step after fine-tuning can lead to the removal of crucial weights resulting in lower performance.

This work aims to mitigate the parameter interference issue in MNMT without adding extra parameters and overcome the aforementioned two limitations in Lin et al. (2021). We propose gradient-based gradual pruning for language-specific MNMT. More specifically, a multilingual base model is first trained, and subsequently, the model is simultaneously fine-tuned and pruned on each language pair. Conducting finetuning and pruning concurrently allows the model to adapt and optimize the parameters while reducing the model size. Instead of the widely used magnitude scores, we opt for the gradient-based scores as the pruning criterion. The percentage of pruned weights is gradually increased from zero to the target level throughout the pruning process. By optimizing the pruning criterion and schedule, we strive to identify optimal sub-networks and limit the parameter interference. Lastly, the resulting sub-networks are integrated into the MNMT model through a final training phase in a language-aware manner.

A large set of experiments is conducted on IWSLT and WMT datasets, showing that our approach leads to a substantial performance gain of 2.06 BLEU on IWSLT and 1.41 BLEU on WMT. Our contributions can be summarized as follows:

- Our method leads to a significant boost in medium- and high-resource languages and also a reasonable improvement in low-resource languages, suggesting its effectiveness in alleviating parameter interference.

- We provide a comprehensive study of various pruning criteria and schedules in searching optimal sub-networks in the multilingual translation scenario.

- We provide additional analyses of our method by studying the contribution of various sub-layers to the overall performance and exploring the relationship between language-specific sub-networks and language families.

## 2 Related Work

Standard multilingual neural machine translation systems translate between multiple languages with a unified model. The model can be jointly trained on multiple language pairs by prepending a special token to the source sentence, informing the model about the desired target language (Johnson et al., 2017). Although fully sharing parameters across languages and joint training can enhance knowledge transfer, MNMT suffers from parameter interference and the lack of language-specific parameters for capturing language-specific information, resulting in performance degradation, especially in high-resource language pairs. Various previous works have explored the idea of partially sharing parameters across languages while allowing for language-specific parameters. Sachan and Neubig (2018) investigates different parameter sharing strategies. Blackwood et al. (2018) compares several methods of designing language-specific attention module. Zhang et al. (2021) studies when and where language-specific capacity matters.

Additionally, another widely recognized technique, adapter-based, has attracted substantial interest in recent years. Adapter-based approaches (Bapna and Firat, 2019; Zhu et al., 2021; Baziotis et al., 2022) inject additional lightweight language-specific modules for different language pairs to capture language-specific information. Although effective, these methods increase the parameters for each language pair and thus result in an inference speed decrease. On the contrary, our work introduces no extra parameters to the model and thus has negligible to no impact on the inference speed. For more details, see Appendix E.

To avoid the inference speed decrease, several works (Dong et al., 2015; Purason and Tättar, 2022; Pfeiffer et al., 2022; Pires et al., 2023) explore designing language-aware modules inside the MNMT model. This way, the model can capture language-specific information without sacrificing the inference speed. For instance, Dong et al. (2015) employs a single shared encoder across all languages but a unique decoder for each target language. However, the model can suffer from the parameter exploration issue as the number of languages in the multilingual model increases. In contrast, our approach maintains a consistent number of total parameters, irrespective of the number of languages.

Recent works allocate language-specific parameters by extracting a unique sub-network for each

language pair within the multilingual model, which avoids introducing additional parameters to the model. Different approaches are explored for sub-network extraction, such as Taylor expansion (Xie et al., 2021), parameter differentiation (Wang and Zhang, 2022), and model pruning (Lin et al., 2021). In this paper, we focus on model pruning because it showed to be more effective and results in promising performance.

In machine learning, model pruning is widely used to remove redundant weights from a neural network while preserving important ones to maintain accuracy (Han et al., 2015, 2016; Frankle and Carbin, 2019; Liu et al., 2019; Sun et al., 2020). Two key components of model pruning are the *pruning criterion*, which determines the relative importance of weights and which weights to prune, and the *pruning schedule*, which defines the strategy how the target pruning ratio is achieved throughout the pruning process. In terms of *pruning criteria*, magnitude pruning (Han et al., 2015, 2016), which removes weights with low absolute values, is the most widely used method for weight pruning. A recent work on large language models (Sanh et al., 2020) proposed movement pruning, which scores weights by using the accumulated product of weight and gradient and removes those weights shrinking toward zero. We refer to this approach as gradient-based pruning in this work. Regarding *pruning schedules,* two commonly considered options are one-shot pruning (Frankle and Carbin, 2019) and gradual pruning (Zhu and Gupta, 2017). While one-shot pruning removes the desired percentage of weights in a single step after the completion of finetuning, gradual pruning incrementally removes weights, starting from an initial pruning ratio (often 0) and gradually progressing towards the target pruning ratio.

In MNMT, the most similar technique to our approach for extracting sub-networks through model pruning has been proposed by Lin et al. (2021). In their work, sub-networks are searched through magnitude one-shot pruning, which means the model is pruned based on the magnitude of weights in a single step after the completion of fine-tuning. Different from their method, we opt for the gradient-based pruning criterion, which is proved to be more effective than magnitude pruning in a transfer learning context. Besides, in contrast to pruning in one single step (Lin et al., 2021), we gradually increase the ratio from zero to the target value during fine-

---

**Algorithm 1** Gradient-based Gradual pruning for MNMT

1: **Input:** $N$ bilingual corpora data $D_{all} = (D_{s_1 \to t_1}, D_{s_2 \to t_2}, ... D_{s_N \to t_N})$, language pairs $pairs = \{s_1 \to t_1, s_2 \to t_2, ..., s_N \to t_N\}$
2:   // Phase 1: pretrain a multilingual model
3:   Training a multilingual model base: $\theta_0$
4:   // Phase 2: extract sub-network for each pair
5: **for** each pair $s_i \to t_i$ in $pairs$ **do**
6:     **if** $step < T_1$ **then**
7:       // Finetune without pruning
8:       Fine-tune and keep $prune\_ratio = 0$
9:     **else if** $step < (T_1 + T_2)$ **then**
10:       Calculate gradient-based importance scores as shown in Eq. (2), and rank the scores.
11:       Calculate the current pruning ratio according to Eq. (4) and prune the weights with the lowest scores accordingly.
12:     **else**
13:       Finetune with target ratio until converge.
14:     **end if**
15:     Extract pruning mask $\mathbf{M}_{s_i \to t_i}$
16:     Extract sub-network $\theta_{s_i \to t_i} = \theta_0 \odot \mathbf{M}_{s_i \to t_i}$
17: **end for**
18:   // Phase 3: structure-aware joint training
19: $\theta_{All} = (\theta_{s_1 \to t_1}, \theta_{s_2 \to t_2}, ..., \theta_{s_N \to t_N})$
20: **while** $\theta_{All}$ not converge **do**
21:     **for** each pair $s_i \to t_i$ in $pairs$ **do**
22:       Further training $\theta_{s_i \to t_i}$ on $D_{s_i \to t_i}$
23:     **end for**
24: **end while**

---

tuning, allowing the model to self-correct and recover from previous choices. To the best of our knowledge, this is the first study to explore the effectiveness of gradient-based gradual pruning in the multilingual translation context.

## 3 Methodology

Our approach includes three main phases. In the first one (Phase 1 in Algorithm 1), a multilingual model is initially trained using the parameter log-likelihood loss (see Sec. 3.1). Subsequently, the multilingual base model is simultaneously fine-tuned and pruned on each language pair (Phase 2 in Algorithm 1). In this phase, we adopt the gradient-based pruning criterion as the scoring function for each weight and the gradual pruning schedule to identify sub-network masks for language pairs (see Sec. 3.2). The extracted masks are then jointly used during the final training (Phase 3 in Algorithm 1) as described in Sec. 3.3.

### 3.1 Multilingual Neural Machine Translation

In this work, we adopt the multilingual Transformer (Vaswani et al., 2017) as the backbone of our approach. Following Lin et al. (2021), we use a unified model for multilingual NMT by adding two

special language tokens to indicate source and target languages. Given a set of $N$ bilingual corpora $D_{all} = (D_{s_1 \to t_1}, D_{s_2 \to t_2}, ...D_{s_N \to t_N})$, the multilingual model is jointly trained over the set of all $N$ parallel training corpora. The objective is to minimize the parameter log-likelihood of the target sentence given the source sentence over all corpora. The training loss is formulated as follows:

$$L_{\text{MT}} = -\sum_{i=1}^{N}\sum_{j=1}^{J} \log p(y_{i,j}|y_{i,<j}, X_i, \theta) \quad (1)$$

where $X_i = (x_{i,1}, x_{i,2}, ..., x_{i,I})$ and $Y = (y_{i,1}, y_{i,2}, ..., y_{i,J})$ represent the source and target sentences of one sentence pair in the parallel corpus $D_{s_i \to t_i}$ respectively, with source sentence length $I$ and target sentence length $J$, and special tokens omitted. The index of the current target word is denoted by $j$, which ranges from 1 to the sentence length $J$. $\theta$ represents the model parameters.

### 3.2 Identify Sub-networks Via Pruning

Once the MNMT model is trained, sub-networks are identified by applying our pruning approach.
**Gradient-based pruning criterion.** Inspired by Sanh et al. (2020), we first learn an importance score for each weight in the weight matrix targeted for pruning, and then prune the model based on these importance scores during the simultaneous finetuning and pruning process. The importance scores can be represented as follows [1]:

$$S_{i,j}^{(T)} = -\sum_{t<T} \left( \frac{\partial L}{\partial W_{i,j}} \right)^{(t)} W_{i,j}^{(t)} \quad (2)$$

where $\frac{\partial L}{\partial W_{i,j}}$ is the gradient of loss $L$ with respect to $W_{i,j}$ in a generic weight matrix $\mathbf{W} \in R^{M \times N}$ of the model, $T$ denotes the number of performed gradient updates, $S_{i,j}^{(T)}$ denotes the importance score of weight $W_{i,j}$ after $T$ updates.

After scoring each weight using Eq. (2) and ranking the score values, we prune the weights having importance scores among the $v\%$ (pruning ratio) lowest, regardless of the absolute score values. To this end, a binary mask matrix $\mathbf{M} \in \{0,1\}^{M \times N}$ based on the importance scores is calculated as:

$$M_{i,j} = \begin{cases} 0 & S_{i,j} \text{ among } v\% \text{ lowest scores} \\ 1, & \text{otherwise} \end{cases} \quad (3)$$

[1]For detailed information please refer to Appendix G.

Weights with scores among the lowest $v\%$ are assigned a value of 0 in the binary mask matrix and pruned, while the other weights are assigned a value of 1 in the mask and retained.

We generate a mask for each matrix that is targeted for pruning in the model and extract a unique sub-network for each language pair: $\theta_{s_i \to t_i} = \theta_0 \odot \mathbf{M}_{s_i \to t_i}$. Where $\odot$ denotes the Hadamard product, $s_i \to t_i$ is the language pair, $\mathbf{M}_{s_i \to t_i} \in \{0, 1\}^{|\theta|}$ represents the mask of the entire model for pair $s_i \to t_i$, $\theta_0$ denotes the initial model, and $\theta_{s_i \to t_i}$ represents the corresponding sub-network for the pair $s_i \to t_i$.

**Gradual pruning schedule.** In this work, the pruning ratio ($v\%$ in Eq. (3)) is gradually increased from 0 to the target value $R_p$ through a three-stage process, similar to Zhu and Gupta (2017). In the first stage spanning $T_1$ training steps, the model remains unpruned with a pruning ratio of 0. In the second stage, which lasts for $T_2$ training steps, the pruning ratio gradually increases from 0 to the predefined threshold $R_p$. In the third stage, the pruning ratio remains constant at the target pruning ratio $R_p$. This strategy is formalized as follows:

$$R_t = \begin{cases} 0 & t < T_1 \\ R_p - R_p \left(1 - \frac{t-T_1}{T_2}\right)^3 & T_1 < t < (T_1 + T_2) \\ R_p & \text{otherwise} \end{cases}$$
$$(4)$$

where $t$ represents the current training step, $R_t$ represents the pruning ratio at step $t$, $R_p$ represents the preset target pruning ratio, $T_1$ and $T_2$ represent the total steps of stage 1 and stage 2, respectively.

### 3.3 Joint Training

Once the sub-networks $\theta_{s_i \to t_i}, i = 1, ..., N$ for all language pairs are obtained, the multilingual base model $\theta_0$ is further trained with language-aware data batching and structure-aware model updating. For this purpose, batches of each language pair are randomly grouped from the language-specific bilingual corpus (Lin et al., 2021). Given a pair $s_i \to t_i$, batches $B_{s_1 \to t_1}$ for this pair are randomly grouped from the bilingual corpus $D_{s_i \to t_i}$. Importantly, each batch contains samples from a single language pair, which differs from standard multilingual training where each batch can contain fully random sentence pairs from all language pairs. The model is iteratively trained on batches of all language pairs until convergence. During back-

propagation, only parameters in the sub-network of the corresponding language pair are updated. During inference, the sub-network corresponding to the specific language pair is utilized to generate predictions. Notably, the sub-networks of all language pairs are accommodated in a single model, without introducing any additional parameters.

# 4 Experiment settings

## 4.1 Data

We run our experiments using two collections of datasets (IWSLT and WMT) having different language coverage and training data sizes. Due to its limited dimensions, the IWSLT data is used to run a deeper analysis of our method while WMT is used to further simulate real-world imbalanced dataset scenarios.

**IWSLT.** We collect 8 English-centric language pairs for a total of 9 languages from IWSLT2014 (Cettolo et al., 2014), with corpus size ranging from 89K to 169K, as shown in Appendix A.1. We first tokenize data with moses scripts (Koehn et al., 2007)[2], and further learn shared byte pair encoding (BPE) (Sennrich et al., 2016), with a vocabulary size of 30K, to preprocess data into sub-word units. To balance the training data distribution, low-resource languages are oversampled using a temperature of T=2 (Arivazhagan et al., 2019).

**WMT.** We collect another dataset comprising 19 languages in total with 18 languages to-and-from English from previous years' WMT (Barrault et al., 2020). The corpus sizes range from very low-resource (Gu, 9K) to high-resource (Fr, 37M). More detailed information about the train, dev, and test datasets is listed in Appendix A.2. We apply the shared byte pair encoding (BPE) algorithm using SentencePiece (Kudo and Richardson, 2018) to preprocess multilingual sentences, with a vocabulary size of 64K. Since the data is extremely imbalanced, we apply oversampling with a larger temperature of T=5. We categorize the language pairs into three groups based on their corpus sizes: low-resource (<1M), medium-resource ($\geq$ 1M and <10M), and high-resource ($\geq$10M).

## 4.2 Model Settings

In our experiments, we adopt models of different sizes to adjust for the variation in dataset sizes. Given the smaller size of the IWSLT benchmark,

we opt for Transformer-small following Wu et al. (2019). For the WMT experiment, we choose Transformer-base. For more training details please refer to Appendix B. To have a fair comparison with Lin et al. (2021), our method is applied to two linear sub-layers: attention and feed-forward.[3]

## 4.3 Terms of Comparison

We compare our method with two well-known and adopted technologies: the adapter-based (Bapna et al., 2019) and the method utilizing a shared encoder but separate decoders (Dong et al., 2015).

- *Adapter.128*, *Adapter.256*, and *Adapter.512* - Inject a unique lightweight adapter to the shared model for each language pair with varying bottleneck dimensions 128, 256, and 512, respectively (Bapna et al., 2019). Following Pires et al. (2023), we train the entire model, including all adapters, from scratch for training stability.

- *SepaDec* - Use a shared encoder for all language pairs and a separate decoder for each target language (Dong et al., 2015).

As described in Section 3, our *GradientGradual* approach employs a gradient-based pruning criterion, and pruning occurs during finetuning, with the pruning ratio gradually increasing. To investigate the effectiveness of our pruning strategy for sub-network extraction in multilingual translation, we further compare our method with three pruning-based variants by combining different pruning criteria and schedules.

- *MagnitudeOneshot (LaSS)* [4] - Extract sub-networks through pruning: the pruning criterion is the magnitude values of the weights; the pruning is performed in one step after the completion of finetuning (Lin et al., 2021).

- *MagnitudeGradual* - Extract sub-networks through pruning: the pruning criterion is the magnitude values of weights; the pruning ratio increases gradually during finetuning.

- *GradientOneshot* - Extract sub-networks through pruning: the pruning criterion is

---

[2]https://github.com/moses-smt/mosesdecoder/blob/master/scripts/tokenizer/tokenizer.perl

[3]Although our analysis is limited to attention and feed-forward sub-layers, our algorithm can be used to prune other components of the model, e.g. the embedding matrixes.

[4]In the original paper, this approach is referred to as LaSS. In the following sections of this paper, we will use *MagnitudeOneshot* for simplicity.

| Lang | Fa | Pl | Ar | He | Nl | De | It | Es | Average |
|---|---|---|---|---|---|---|---|---|---|
| Size | 89K | 128k | 140K | 144K | 153K | 160K | 167K | 169K | |
| Baseline | 17.54 | 17.25 | 21.31 | 29.39 | 32.11 | 30.09 | 30.02 | 36.18 | 26.74 |
| *Adapter.128* | +0.20 | **+1.46** | +1.32 | +1.56 | +1.62 | +1.34 | +1.36 | +1.82 | +1.33 |
| *Adapter.256* | +0.06 | +1.44 | +1.53 | +1.56 | +1.43 | +1.52 | +1.53 | +1.70 | +1.34 |
| *Adapter.512* | -0.23 | +1.39 | +1.61 | +2.09 | +1.67 | +1.81 | +2.01 | +2.01 | +1.54 |
| *SepaDec* | -0.38 | +0.79 | +0.88 | +0.84 | +0.75 | +1.23 | +1.12 | +0.73 | +0.74 |
| *MagnitudeOneshot* (LaSS) | +0.04 | +0.26 | +0.68 | +0.65 | +0.56 | +0.49 | +0.72 | +0.60 | +0.50 |
| *GradientGradual* (Ours) | **+1.13** | +0.67 | **+2.24** | **+2.8** | **+2.56** | **+2.28** | **+2.19** | **+2.60** | **+2.06** |
| *MagnitudeGradual* | +0.43 | +0.20 | +2.22 | +2.36 | +2.07 | +1.92 | +2.02 | +2.24 | +1.68 |
| *GradientOneshot* | +0.73 | +0.32 | +2.08 | +2.44 | +2.14 | +2.08 | +1.88 | +2.14 | +1.73 |

Table 1: Average En ↔ X BLEU score gain of the sub-network extraction methods on the IWLST dataset.

gradient-based; pruning is performed in one step after the completion of finetuning.

## 4.4 Evaluation

We report the tokenized BLEU (Papineni et al., 2002) on individual languages on the IWSLT dataset, and average tokenized BLEU of low-, medium-, and high-resource groups on the WMT dataset using the SacreBLEU tool (Post, 2018). We provide detailed results on the WMT dataset in Appendix D.1. In addition, we show the results of win ratio based on tokenized BLEU in Appendix D.2, and the results of COMET (Rei et al., 2020) and chrF (Popović, 2015) scores in Appendix D.3.

## 5 Experiment Results on IWSLT

This section shows the results of our approach on the IWSLT dataset, along with a comparative analysis against the MNMT baseline and prior research works. We also provide a comprehensive analysis of our method from various perspectives. [5]

## 5.1 Main Results

In Table 1, we first report the performance of multilingual baseline, adapter-based approaches (*Adapter.128*, *Adapter.256*, and *Adapter.512* with bottleneck dimensions 128, 256, and 512) and *SepaDec* method, with a dedicated decoder for each target language. On average, *Adapter.128*, *Adapter.256*, and *Adapter.512* achieve 1.33, 1.34, and 1.54 BLEU score gains, respectively. *SepaDec* clearly underperforms adapter-based methods yet exhibits a smaller gain of 0.74 BLEU score points over the baseline. The above approaches alleviate the parameter interference issue by introducing additional language-specific parameters to the model and outperform the baseline across

most languages. The consistent improvements in average scores confirm the necessity of language-specific parameters in the MNMT model. Subsequently, we report the results of the pruning-based approach *MagnitudeOneshot*, which extracts sub-networks for language pairs through magnitude pruning in one step. *MagnitudeOneshot* obtains a 0.50 BLEU score gain, demonstrating the potential of designing language-specific parameters and mitigating parameter interference without increasing the parameter count. Furthermore, our proposed *GradientGradual* approach, which leverages gradient-based information as the pruning criterion and gradually increases the pruning ratio from 0 to the target value, delivers a substantial improvement of 2.06 BLEU scores over the baseline model and achieves the best performance among all approaches, suggesting the effectiveness of our method in mitigating parameter interference in the MNMT model.

## 5.2 Pruning criteria and schedules

In the last two rows of Table 1 we further explore the individual impact of two key factors that contribute to the performance improvement separately, namely the gradient-based pruning criterion and gradual pruning schedule. *MagnitudeGradual* leads to a 1.68 BLEU score improvement over the baseline model and a 1.18 BLEU score improvement over *MagnitudeOneshot*, separately. *GradientOneshot* leads to a 1.73 BLEU score gain over baseline and a 1.23 BLEU score gain over *MagnitudeOneshot*. The results demonstrate that both the gradient-based pruning criterion and gradual pruning schedule have the potential to significantly improve multilingual translation performance when implemented separately and to surpass the performance of *MagnitudeOneshot*. Nevertheless, the maximum average improvement is obtained when the gradient-based pruning criterion and gradual

---

[5]Additional analysis of parameter count, disk storage, and inference speed among different approaches is provided in Appendix E .

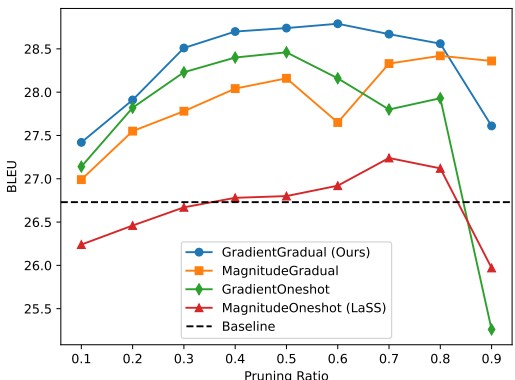

Figure 1: Average BLEU score of all language pairs across 4 methods with pruning ratio [0.1:0.9].

pruning schedule are combined, as demonstrated in our proposed *GradientGradual* approach.

### 5.3 Robustness with respect to pruning ratios

Results presented in Table 1 demonstrate that all 4 methods with different pruning criteria and schedules lead to varying levels of performance improvement. To gain a more comprehensive understanding of the relationship between performance and pruning ratios, we further visualize the average performance of all language pairs across the 4 methods with different pruning ratios. Results in Figure 1 show that the optimal performance of these 4 methods is obtained at slightly different pruning ratios. In addition, although *MagnitudeGradual* and *GradientOneshot* yield a considerably higher performance gain (around 1.7 BLEU) than *MagnitudeOneshot* (0.5 BLEU) at their optimal pruning ratios, they both show instability across pruning ratios. Specifically, *GradientOneshot* suffers from a significant performance drop from middle to high pruning ratios, and *MagnitudeGradual* experiences an unexpected performance drop at the specific pruning ratio of 0.6. In contrast, *GradientGradual* demonstrates a more robust behavior across a wide range of pruning ratios, consistently outperforming the other methods except for *MagnitudeGradual* at a very high pruning ratio of 0.9. These results further verify the effectiveness and robustness of our *GradientGradual* approach.

### 5.4 Which sub-layer matters?

In this work, our approach is applied to attention and feed-forward sub-layers. To better understand where the parameter interference is more severe and where language-specific parameters are essen-

tial, we perform ablation experiments by applying our approach to the attention and feed-forward sub-layers separately. The results in Table 2 show that applying our approach to feed-forward sub-layers yields a limited average performance gain (+0.49 BLEU) while applying it to attention sub-layers leads to a notable gain (+1.22 BLEU), suggesting parameters in attention sub-layers are more language-specific. This finding aligns with the previous work (Clark et al., 2019), which shows that specific attention heads specialize in distinct aspects of linguistic syntax. Given the unique syntax patterns in different languages, parameters in attention sub-layers are possibly more language-specific and suffer more severe parameter interference. Consequently, applying our approach to attention sub-layers yields a notable gain. However, the largest average performance gain (+2.06 BLEU) is achieved when applying our approach to both attention and feed-forward sub-layers. Our results suggest that parameter interference exists in both sub-layers, but is more severe in attention sub-layers.

### 5.5 Similarity Scores and Phylogenetic Tree

To gain deeper insights into the effectiveness and interpretability of our method and to evaluate its capability to extract high-quality language-specific sub-networks, we compute the similarity of masks obtained with our approach, and we use these similarities to reconstruct the phylogenetic trees of languages. We present the results of En→X language pairs in this section and the results of X→En, which exhibit similar patterns as En→X in Appendix F.

Similarity scores are determined by the proportion of shared parameters between two language pairs (Lin et al., 2021), which can be obtained by dividing the number of shared "1" values in two masks by the number of "1" in the first mask, as illustrated in the equation below:

$$\mathbf{Sim}(\mathbf{M1}, \mathbf{M2}) = \frac{\|\mathbf{M1} \odot \mathbf{M2}\|_0}{\|\mathbf{M1}\|_0}$$

where $\|\cdot\|_0$ is $L_0$ norm, $\mathbf{M1}$ and $\mathbf{M2}$ represent binary masks of two language pairs, $\|\mathbf{M1} \odot \mathbf{M2}\|_0$ represents the number of shared 1, i.e., the number of shared parameters, in these two language pairs. Intuitively, languages within the same family share a higher linguistic similarity, implying an increased likelihood of shared parameters and higher similarity scores. Conversely, languages that are

| Lang | Fa | Pl | Ar | He | Nl | De | It | Es | Average |
|---|---|---|---|---|---|---|---|---|---|
| Size | 89K | 128k | 140K | 144K | 153K | 160K | 167K | 169K | |
| Baseline | 17.54 | 17.25 | 21.31 | 29.39 | 32.11 | 30.09 | 30.02 | 36.18 | 26.74 |
| Attn | +0.45 | +0.09 | +1.39 | +1.95 | **+1.4** | **+1.33** | +1.57 | +1.58 | +1.22 |
| Ff | +0.1 | -0.01 | +0.79 | +0.74 | +0.41 | +0.63 | +0.71 | +0.54 | +0.49 |
| AttnFf | **+1.13** | **+0.67** | **+2.24** | **+2.8** | +1.13 | +0.67 | **+2.24** | **+2.8** | **+2.06** |

Table 2: Sub-layer Ablation Results. Baseline denotes the multilingual Transformer model. Attn denotes applying our approach to attention sub-layers only. Ff denotes the approach applied to the feed-forward sub-layers only. AttnFf represents applying our approach to both the attention and feed-forward sub-layers.

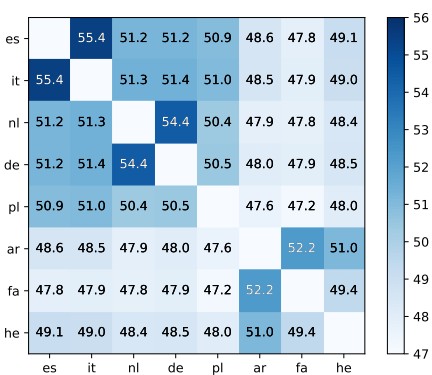

Figure 2: Similarity scores (%) of En→X language pairs.

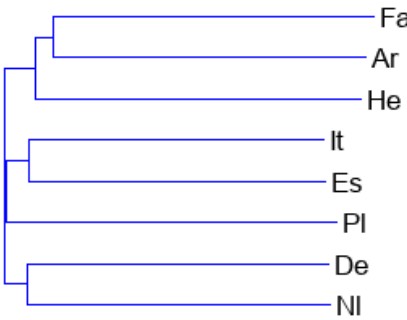

Figure 3: The built tree positively correlates with language families. It and Es are in the Romance branch; De and Nl are in the Germanic branch. Fa, Ar, and He are similar Afro-Asiatic languages; with Fa and Ar written in Arabic script.

linguistically distant from one another tend to possess more distinct language-specific characteristics, which implies lower similarity scores.

Figure 2 shows the similarity scores of En→X language pairs. Italian (It) and Spanish (Es) obtained the highest score, followed by German (De) and Dutch (Nl). The third highest score is obtained by Arabic (Ar) and Farsi (Fa). In addition, we noticed that within the European or Afro-Asiatic language groups, similarity scores between languages are relatively high, while lower scores are often observed when comparing a language from the European group to one from the Afro-Asiatic group. The results demonstrate that similarity scores obtained with our method are highly positively correlated with language family clustering (see Table 14 in Appendix F). This implies the capability of our *GradientGradual* approach to generate high-quality language-specific sub-network.

Furthermore, to quantify the proportion of language-specific parameters and construct a phylogenetic tree of languages, we compute language distance scores as: $\mathbf{Dis}(\mathbf{M1}, \mathbf{M2}) = 1 - \mathbf{Sim}(\mathbf{M1}, \mathbf{M2})$.

With these scores calculated between every two language pairs, the phylogenetic tree is constructed according to a weighted least-squares criterion

(Makarenkov and Leclerc, 1999) using the T-REX tool (Boc et al., 2012)[6]. Figure 3 demonstrates the constructed phylogenetic tree is highly aligned with the language families shown in Table 14 in Appendix F, confirming the capability of our approach to extract high-quality language-specific sub-networks.

## 6 Experiment Results on WMT

On the highly imbalanced WMT dataset with the Transformer-base model, we compare our method with the multilingual baseline and the prior work *MagnitudeOneshot*, which shares the same philosophy as our method: both our approach and *MagnitudeOneshot* introduce no additional parameters to the model and have no negative impact on the inference speed. The results in Table 3 show that our method outperforms both the baseline and the *MagnitudeOneshot* across low-, medium-, and high-resource language pairs, reconfirming the effectiveness of our approach. More specifically, we observe an average improvement of 0.91 BLEU on low-resource, 1.56 BLEU on medium-resource, and 1.67 BLEU on high-resource language pairs

---

[6]http://www.trex.uqam.ca/index.php?action=trex

| Model | Low | Med | High | All |
|---|---|---|---|---|
| Baseline | 14.01 | 17.54 | 24.16 | 18.76 |
| *MagnitudeOneshot* | +0.70 | +0.85 | +0.99 | +0.85 |
| *GradientGradual* (Ours) | **+0.91** | **+1.56** | **+1.67** | **+1.41** |

Table 3: Average BLEU improvement of Low ($< 1M$), Medium ($\geq 1M$ and $< 10M$) and High ($\geq 10M$) resource language pairs over baseline on WMT dataset.

over the baseline. These results shed light on three aspects. Firstly, the performance improvement becomes larger when the number of training resources increases, which validates findings in previous research works, suggesting that high-resource language pairs tend to be more negatively affected by the parameter interference issue in a unified multilingual model. As a result, mitigating the interference issue can result in more notable enhancements for high-resource languages. Secondly, while previous works suggest that low-resource languages often benefit from knowledge transfer, our results suggest that parameter interference also harms the performance of low-resource languages. By mitigating the interference issue, the performance of low-resource languages can also be improved. Finally, the consistent improvement of our method compared to *MagnitudeOneshot* across language groups of all resource sizes suggests that our gradient-based gradual pruning approach is more effective in identifying optimal sub-networks and mitigating the parameter interference issue in the multilingual translation scenario.

## 7    Conclusion

In a standard MNMT model, the parameters are shared across all language pairs, resulting in the parameter interference issue and compromised performance. In this paper, we propose gradient-based gradual pruning for multilingual translation to identify optimal sub-networks and mitigate the interference issue. Extensive experiments on the IWSLT and WMT datasets show that our method results in large gains over the normal MNMT system and yields better performance and stability than other approaches. Additionally, we observe that the interference issue can be more severe in attention sublayers and it is possible to reconstruct a reliable phylogenetic tree of languages using the language-specific sub-networks generated by our approach. All the experiments confirm the effectiveness of our approach and the need for better training strategies to improve MNMT performance.

## Limitations

### Modelling

In this work, we explore our approach in the English-centric multilingual machine translation setting. However, we believe that the effectiveness of our method extends to the real-world non-English-centric scenario. We will explore this direction in future work.

Regarding model capacity, we adopt Transformer-small and Transformer-base for IWSLT and WMT experiments, respectively, to reduce the training cost. Given the extensive coverage of languages in the WMT dataset, using an even larger model like Transformer-big (Vaswani et al., 2017) may further boost the overall translation performance, especially in high-resource directions, but we do not expect any difference in the relative quality of the tested methods (*GradientGradual* > *MagnitudeOneshot* > Baseline). This has been confirmed when moving from Transformer-small to Transformer-base in our experiments.

Another limitation of our work is that the current version of our algorithm applies the same pruning ratio to all language pairs and, hence, prunes the same percentage of parameters for each language. However, language pairs with different sizes of available training data may have different optimal pruning ratios. For example, high-resource language pairs may need smaller pruning ratios and preserve sufficient parameters to process and capture more complex information in the abundant data. To potentially improve the gains and have a method able to differently behave with different data conditions, future work could explore a method to automatically identify appropriate pruning ratios for different languages.

### Training

In this work, we aim to search for a sub-network for each language pair within the multilingual model to mitigate the parameter interference issue and improve the performance. While our approach avoids introducing additional parameters, the 3-phase training process (training a base model, searching for sub-networks, and joint training) introduces additional complexity to the training pipeline compared to the standard end-to-end multilingual model training, and demands computational resources for the sub-network searching phase.

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

# A  Datasets details

In this section, we provide detailed data information of our experiments.

## A.1  IWSLT dataset

Table 4 provides detailed information about the IWLST data.

| ISO | Language | Family | Script | Size (K) |
|-----|----------|--------|--------|----------|
| fa | Farsi | Iranian | Arabic | 89 |
| ar | Arabic | Arabic | Arabic | 140 |
| he | Hebrew | Semitic | Hebrew | 144 |
| it | Italian | Romance | Latin | 167 |
| es | Spanish | Romance | Latin | 169 |
| pl | Polish | Slavic | Latin | 128 |
| nl | Dutch | Germanic | Latin | 153 |
| de | German | Germanic | Latin | 160 |

Table 4: Statistics of IWSLT data with similar languages organized into groups.

## A.2  WMT dataset

Table 5 provides detailed information about the WMT data.

# B  Model details

In this section, we provide detailed model information in our experiments.

**IWSLT.** Given the small scale of the data in the IWSLT dataset, we adopt the Transformer-small architecture with 4 attention heads: $L = 6$, $d = 512$, $n_{head} = 4$ and $d_{ff} = 1024$.

**WMT.** For the WMT experiment, we adopt a Transformer-base architecture with 8 attention heads: $L = 6$, $d = 512$, $n_{head} = 8$ and $d_{ff} = 2048$.

# C  Training details

As shown in section 3, our approach includes three phases: training a multilingual base model (Phase 1), identifying sub-networks through pruning (Phase 2), and joint training (Phase 3). In this section, we provide the details of the hyperparameters of these 3 phases in our experiments.

## C.1  IWSLT

In Phase 1, we train the multilingual base model with the same set of hyper-parameters as in Lin et al. (2021). More specifically, we optimize parameters with Adam (Kingma and Ba, 2015) ($\beta 1 = 0.9$, $\beta 2 = 0.98$), a learning rate schedule of (5e-4,4k), dropout of 0.1 and label smoothing of 0.1. The max tokens per batch is set to 262144. The maximum update number is set to 160K with a checkpoint saved

| ISO | Language | Family | Script | Train | Valid | Test | Size |
|---|---|---|---|---|---|---|---|
| gu | Gujarati | Indo-Aryan | Gujarati | WMT19 | newsdev19 | newstest19 | 7.5K |
| ta | Tamil | Dravidian | Tamil | WMT20 | newsdev20 | newstest20 | 605K |
| kk | Kazakh | Turkic | Cyrillic | WMT19 | newsdev19 | newstest19 | 102K |
| tr | Turkish | Turkic | Latin | WMT16 | newsdev16 | newstest16 | 198K |
| ro | Romanian | Romance | Latin | WMT16 | newsdev16 | newstest16 | 596K |
| es | Spanish | Romance | Latin | WMT13 | newstest12 | newstest13 | 12.9M |
| fr | French | Romance | Latin | WMT14 | newstest13 | newstest14 | 37M |
| ps | Pashto | Iranian | Arabic | WMT20 | wikipedia | newstest20 | 1M |
| fi | Farsi | Uralic | Latin | WMT16 | newstest15 | newstest16 | 2M |
| lv | Latvian | Baltic | Latin | WMT17 | newsdev17 | newstest17 | 2.1M |
| et | Estonian | Uralic | Latin | WMT18 | newsdev18 | newstest18 | 2.1M |
| lt | Lithuanian | Baltic | Latin | WMT19 | newsdev19 | newstest19 | 3.4M |
| ru | Russian | Slavic | Cyrillic | WMT16 | newstest15 | newstest16 | 1.4M |
| cs | Czech | Slavic | Latin | WMT14 | newstest13 | newstest14 | 11M |
| pl | Polish | Slavic | Latin | WMT20 | newsdev20 | newstest20 | 11M |
| ja | Japanese | Japonic | Kanji; Kana | WMT20 | newsdev20 | newstest20 | 16.5M |
| zh | Chinese | Chinese | Chinese | WMT17 | newsdev17 | newstest17 | 11.9M |
| de | German | Germanic | Latin | WMT16 | newstest13 | newstest14 | 4.4M |

Table 5: Statistics of WMT data with similar languages organized into groups.

| Lang | Gu | Kk | Tr | Ro | Ta | Average |
|---|---|---|---|---|---|---|
| Size | 7.5K | 102K | 198K | 596K | 605K | |
| | | | En → X translation | | | |
| Baseline | 1.2 | 2.5 | 17.8 | 31.1 | 14.1 | 13.34 |
| MagnitudeOneshot | +0.1 | +1.3 | +1.7 | +0.8 | +3.1 | +1.40 |
| GradientGradual (Ours) | −0.2 | +1.4 | +1.7 | +1.9 | +6.0 | **+2.16** |
| | | | X → En translation | | | |
| Baseline | 0.3 | 9.5 | 21.3 | 26.6 | 15.7 | 14.68 |
| MagnitudeOneshot | +0.4 | −0.5 | −0.4 | +0.2 | +0.3 | **0.00** |
| GradientGradual (Ours) | 0.0 | −2.5 | −0.2 | +0.8 | +0.2 | −0.34 |

Table 6: BLEU score over baseline of each pair in the low-resource group ($< 1M$) on the WMT dataset.

every 500 updates, and the patience for early stop training is set to 30. In Phase 2, we set the max tokens to 16384, and dropout to 0.3. The training steps of 3 stages are set to 4K, 36K, and 40K. The best performance of the final model is achieved with a pruning ratio of 0.6 in this phase. The other settings are as same as in Phase 1. In Phase 3, we keep the same settings as Phase 1, except we apply masks on the model.

## C.2  WMT

In Phase 1, the parameters are as same as in Lin et al. (2021). We train the multilingual base model with Adam (Kingma and Ba, 2015) ($\beta1 = 0.9$, $\beta2 = 0.98$), a learning rate schedule of (5e-4,4k), dropout of 0.1 and label smoothing of 0.1. The max tokens per batch is set to 524288. The maximum update

number is set to 600K with a checkpoint saved every 1K updates, and the patience for early stop training is set to 30. In Phase 2, the max tokens per batch are set to 20K, 40K, 80K, and 160K for languages with training data sizes >10K, >100K, >1M, and >10M. The training steps of 3 stages are set to 4K, 16K, and 20K. The best performance of the final model is achieved with a pruning ratio of 0.2 in this phase. In Phase 3, we keep the same settings as Phase 1, except we apply masks on the model.

## D  Additional evaluation results

### D.1  Detailed BLEU scores on WMT dataset

Considering the relatively extensive number of languages involved in the WMT dataset, we present average scores for low-, medium-, and high-resource

| Lang | Ps | Fi | Lv | Et | Ru | Lt | De | Average |
|---|---|---|---|---|---|---|---|---|
| Size | 1M | 2M | 2.1M | 2.1M | 2.3M | 3.4M | 4.4M | |
| $En \rightarrow X$ translation | | | | | | | | |
| Baseline | 4.1 | 14.7 | 14.1 | 15.1 | 22.2 | 8.6 | 19.9 | 14.10 |
| MagnitudeOneshot | +0.4 | +1.8 | +1.1 | +1.3 | +2.4 | +0.1 | +2.0 | +1.30 |
| GradientGradual (Ours) | +2.6 | +2.3 | +2.0 | +2.3 | +3.5 | 0.0 | +2.9 | **+2.23** |
| $X \rightarrow En$ translation | | | | | | | | |
| Baseline | 9.8 | 22.1 | 18.8 | 22.4 | 28.7 | 17.1 | 27.9 | 20.97 |
| MagnitudeOneshot | 0.0 | +0.4 | +0.4 | +0.5 | +0.5 | +0.3 | +0.7 | +0.40 |
| GradientGradual (Ours) | +1.1 | +0.7 | +0.6 | +1.2 | +1.1 | +0.3 | +1.3 | **+0.90** |

Table 7: BLEU score over baseline of each pair in the medium-resource group (1M–10M) on the WMT dataset.

| Lang | Cs | Pl | Zh | Es | Ja | Fr | Average |
|---|---|---|---|---|---|---|---|
| Size | 11M | 11M | 11.9M | 12.9M | 16.5M | 37M | |
| $En \rightarrow X$ translation | | | | | | | |
| Baseline | 19.4 | 15.9 | 22.7 | 29.4 | 20.0 | 32.6 | 23.33 |
| MagnitudeOneshot | +1.9 | +1.6 | +0.7 | +1.1 | +1.2 | +1.8 | +1.38 |
| GradientGradual (Ours) | +2.7 | +1.6 | +3.0 | +2.0 | +2.3 | +2.50 | **+2.35** |
| $X \rightarrow En$ translation | | | | | | | |
| Baseline | 28.7 | 24.8 | 15.5 | 31.4 | 15.8 | 33.7 | 24.98 |
| MagnitudeOneshot | −0.5 | +0.8 | +1.7 | +0.4 | +0.8 | +0.4 | +0.60 |
| GradientGradual (Ours) | −0.2 | +0.7 | +2.5 | +1.1 | +1.2 | +0.6 | **+0.98** |

Table 8: BLEU score over baseline of each pair in the high-resource group (>10M) on the WMT dataset.

| Lang | Low | Med | High | All |
|---|---|---|---|---|
| WR (%) | 60 | **86** | **83** | **78** |

Table 9: Average win ratios of Low ($< 1M$), Medium ($\geq$ 1M and $<$ 10M) and High ($\geq$10M) resource language pairs of our *GradientGradual* approach over the *MagnitudeOneshot* approach

| Model | COMET | chrF |
|---|---|---|
| Baseline | 0.773 | 51.2 |
| *Adapter.128* | 0.785 | 52.5 |
| *Adapter.256* | 0.785 | 52.2 |
| *Adapter.512* | 0.785 | 52.3 |
| *SepaDec* | 0.774 | 51.6 |
| *MagnitudeOneshot* | 0.775 | 51.5 |
| *GradientGradual* (Ours) | **0.788** | **52.9** |

Table 10: Average COMET and chrF scores across all language pairs of various approaches on the IWSLT dataset.

groups to offer an overview of performance in different data situations in Table 3. For the detailed results, we provide BLEU scores of languages in low, medium, and high resource groups in Table 6, Table 7, and Table 8, respectively.

## D.2   Win ratio results

To gain further insights into the performance of our approach, we present the results of win ratio (WR) based on tokenized BLEU, which denotes the percentage of languages where our approach outperforms the other. We choose the *MagnitudeOneshot* (LaSS) as a strong baseline to compare with. Table 1 shows that our *GradientGradual* approach outperforms *MagnitudeOneshot* across all languages on the IWSLT dataset, i.e., the win ratio is 100%. Table 9 presents the win ratio (WR) results on the more imbalanced WMT dataset. The results show that our approach achieves win ratios of 60%, 86% and 83% on low-, medium-, and high-resource languages, further verifying the superiority of our approach across different data sizes and especially on medium-, and high-resource languages.

## D.3   COMET and chrF scores

We present the averaged COMET and chrF scores of all language pairs on the IWLST dataset in Table 10 and the WMT dataset in Table 11. More specifically, on the IWLST dataset, we report the scores for the multilingual baseline and various existing approaches proposed in prior works, i.e., the Adapter-based approach with different bottleneck dimensions: *Adapter.128*, *Adapter.256*, and *Adapter.512*; *SepaDec*, *MagnitudeOneshot*, as well as our *GradientGradual* method. On the WMT dataset, we show the scores of the multilingual baseline, *MagnitudeOneshot*, and our approach. On both the IWLST and the WMT datasets, our method outperforms other approaches on COMET and chrF metrics.

| Model | COMET | chrF |
|---|---|---|
| Baseline | 0.733 | 44.4 |
| *MagnitudeOneshot* | 0.747 | 45.4 |
| *GradientGradual* (Ours) | **0.755** | **45.9** |

Table 11: Average COMET and chrF scores across all language pairs of various approaches on the WMT dataset.

| Model | BLEU | $|\theta_l|$ | $|\theta_{all}|$ | Disk storage |
|---|---|---|---|---|
| Baseline | 26.74 | 78M | 78M | 298M |
| *Adapter.128* | 28.07 | 79M | 103M | 397M |
| *Adapter.256* | 28.08 | 81M | 128M | 493M |
| *Adapter.512* | 28.28 | 84M | 179M | 685M |
| *SepaDec* | 27.48 | 78M | 478M | 1.8G |
| *MagnitudeOneshot* | 27.24 | 78M | 78M | 497M |
| *GradientGradual* | **28.80** | 78M | 78M | 497M |

Table 12: Comparison of performance, parameter count, and disk storage among different approaches.

# E Additional metrics

In this section, we compare our method with various approaches from the perspectives of total model parameter counts ($|\theta_{all}|$), parameter counts of individual languages ($|\theta_l|$) during inference, disk storage, inference speed, and the average BLEU scores of all language pairs as performance. We present the primary results in Table 12 and detailed inference speed results of our method in Table 13. The results are obtained with the Transformer-small architecture on the IWSLT dataset.

**Parameter count.** As shown in Table 12, adapter-based approaches increase both the total model parameter counts and the parameter counts of individual languages. *SepaDec* keeps parameter counts of individual languages constant but leads to the most significant total model parameter count due to separate decoders. In contrast, without introducing additional parameters, *MagnitudeOneshot* and our *GradientGradual* approach maintain the same parameter counts for both the overall models and individual languages as the multilingual NMT baseline model.

**Disk requirement.** Similar to the parameter count analysis, adapter-based approaches cause different levels of disk storage increase for added parameters depending on the bottleneck dimensions. *MagnitudeOneshot* and our *GradientGradual* approach also require a moderate amount of extra disk stor-

| Model | Ratio | Speed GPU | Speed CPU |
|---|---|---|---|
| Baseline | – | 237.56 ± 2.70 | 59.98 ± 2.73 |
| *GradientGradual* | 0.1 | 235.23 ± 3.65 | 59.57 ± 0.77 |
| *GradientGradual* | 0.2 | 233.93 ± 3.28 | 58.81 ± 4.27 |
| *GradientGradual* | 0.3 | 235.71 ± 1.92 | 58.26 ± 3.34 |
| *GradientGradual* | 0.4 | 236.35 ± 2.71 | 58.04 ± 2.93 |
| *GradientGradual* | 0.5 | 235.94 ± 2.08 | 59.37 ± 1.73 |
| *GradientGradual* | 0.6 | 238.76 ± 2.49 | 58.94 ± 3.28 |
| *GradientGradual* | 0.7 | 240.10 ± 2.62 | 60.17 ± 2.70 |
| *GradientGradual* | 0.8 | 242.55 ± 2.47 | 60.58 ± 2.37 |
| *GradientGradual* | 0.9 | 243.23 ± 2.89 | 61.56 ± 3.99 |

Table 13: Tokens/second comparison of our approach against the baseline on the IWSLT De→En test set.

age for storing indices. However, the most significant disk storage requirement is from *SepaDec* due to the largest number of total parameters.

**Inference speed.** Adapter-based approaches introduce additional parameters during inference. A larger bottleneck dimension of the adapter leads to a greater expansion of parameters, consequently causing a more considerable inference speed decrease. *SepaDec*, on the other hand, maintains the same number of parameters as the baseline during inference and, therefore, has insignificant to no impact on inference speed. Below, we demonstrate that our *GradientGradual* method, has a negligible effect on inference speed with detailed GPU and CPU speed results. [7]

In our approach, the sub-network mask matrices are represented as binary matrices and are obtained during training and directly applied to Attention and Feedforward sub-layers during inference. Compared to the compute-intensive multiplication between dense weight matrices, multiplying a binary mask matrix with a weight matrix on some sub-layers introduces a minimal overhead and, therefore, has a negligible impact on inference speed. In addition, zero elements in the masked weight matrix could result in faster inference speed due to the possibility of avoiding unnecessary arithmetic operations. We report tokens/second on the IWSLT De→En test set in Table 13. The batch size is always set to 1 and the result is averaged over 5 runs. Speed GPU is measured using a single NVIDIA A100 GPU and Speed CPU is measured using a single-threaded Intel(R) Xeon(R) Gold 6330 CPU @ 2.00GHz.

---

[7]The analysis also applies to *MagnitudeOneshot*.

While the best performance of our approach is achieved with pruning ratio set to 0.6, as shown in Figure 1, we provide the inference speed results with the pruning ratio ranging from 0.1 to 0.9 to offer a more comprehensive analysis of inference speed. Although Speed GPU and CPU are different in absolute value, they show a similar pattern across different pruning ratios, and we take the Speed GPU as an example to analyze. The results indicate an improvement in the inference speed of our approach when compared with the multilingual NMT baseline within the pruning ratio range of 0.6 to 0.9, and the highest inference speed is obtained when pruning ratio is set to 0.9. We attribute this speed improvement to the avoidance of unnecessary operations with element 0. Additionally, the inference speed is slightly slower than the baseline, with pruning ratio smaller than 0.6, but not statistically significantly different.

**Performance.** Our approach outperforms *SepaDe* by 1.53 BLEU scores and surpasses the Adapter-based approaches with 128, 256, and 512 bottleneck dimensions by 0.72, 0.73, and 0.51 BLEU scores, respectively. Additionally, compared to *MagnitudeOneshot*, the work most similar to ours, our approach outperforms by 1.53 BLEU scores.

Based on the comprehensive analysis above, our approach is particularly advantageous when prioritizing performance and inference speed is crucial, and some additional disk requirements are considered acceptable.

## F Similarity Scores and Phylogenetic Tree of X→En language pairs

Table 14 reports detailed language family information for the languages in the IWSLT dataset. In particular, it includes the cluster, branch, and script of each language. Languages belonging to the same cluster and branch are expected to be linguistically closer to each other and have relatively high similarity scores.

Figure 4 shows the similarity scores of X→En language pairs. The results demonstrate that the similarity scores of languages belonging to the same cluster are relatively high. Italian (It) and Spanish (Es) obtained the highest score, followed by German (De) and Dutch (Nl). The third highest score is obtained by Arabic (Ar) and Hebrew (He). Besides, the similarity scores of languages between two distinct language clusters are relatively low, with the lowest score obtained by Dutch (Nl), a

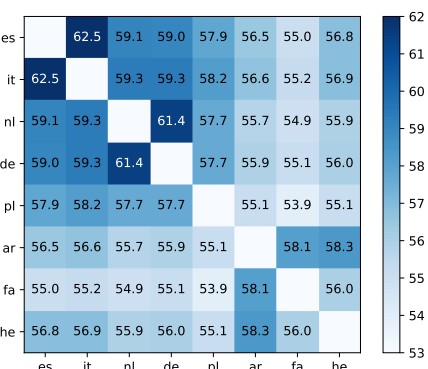

Figure 4: Similarity scores (%) of X→En language pairs. The scores are represented on both the x-axis and y-axis.

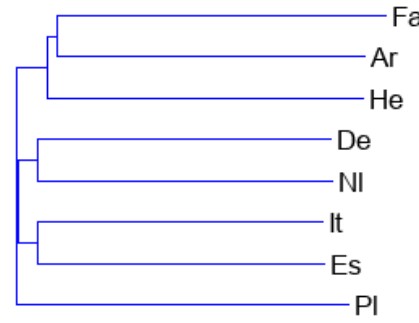

Figure 5: Phylogenetic Tree built with X→En distance scores obtained in our method. The built tree shows a strong positive correlation with language families. It, Es, De, Nl, and Pl are European languages written in Latin; With It, and Es in the Romance branch; De and Nl in the Germanic branch. Fa, Ar, and He are similar Afro-Asiatic languages; with Fa and Ar written

European language, and Farsi (fa), an Afro-Asiatic language. Figure 5 shows the corresponding phylogenetic tree obtained with distance scores.

## G Gradient-based pruning criterion

In this paper, we apply gradient-based pruning to MNMT. This approach uses gradient-based information to identify the language-specific sub-networks. More specifically, to identify which weights to prune in a given weight matrix $\mathbf{W}$, a scoring matrix $\mathbf{S}$ and a binary mask matrix $\mathbf{M}$ are introduced in association with the weight matrix. Each parameter in the score matrix is intended to capture and learn the importance of the corresponding weight, and each element in the binary mask is assigned a value of either 0 or 1 according to whether the corresponding weight is pruned or

| ISO | Language | Cluster | Branch | Script |
|-----|----------|---------|--------|--------|
| fa | Farsi | Afro-Asiatic | Iranian | Arabic |
| ar | Arabic | Afro-Asiatic | Semitic | Arabic |
| he | Hebrew | Afro-Asiatic | Semitic | Hebrew |
| it | Italian | European | Romance | Latin |
| es | Spanish | European | Romance | Latin |
| pl | Polish | European | Slavic | Latin |
| nl | Dutch | European | Germanic | Latin |
| de | German | European | Germanic | Latin |

Table 14: Language clustering of 8 languages in IWSLT

retained. Weights with relatively low scores in the score matrix are considered less important and assigned a value of 0 in the binary mask matrix. Score parameters are learned and updated iteratively during the training process. The scores of all the weights, both pruned and retained weights, are updated. The updating of scores can change the relative importance of different weights and affect their score distribution. This process enables the model to self-correct by allowing pruned weights to come back.

In the following demonstration, we will show how the learned scores are based on gradient information, as depicted in Eq. (2). In the forward pass of the training process, the masking step, where the output is 1 if the input (in this context, the score) is above a threshold and 0 otherwise, is performed after the linear operation. The output of the linear operation and masking can be calculated as $a_i = \sum_{k=1}^{N} W_{i,k} M_{i,k} x_k$. During backpropagation, the gradients of learnable parameters are computed to update these parameters and facilitate the learning process. However, the masking step, introduces a non-differentiable behavior at the threshold point. Besides, the constant output of 1 or 0 results in a gradient of 0 everywhere it is defined. This can lead to the so-called "vanishing gradient" issue, which arises when the gradients become very small or vanish at some point during backpropagation. As a result, the flow of useful gradient information is hindered, making it difficult to train the model effectively. Thanks to Bengio et al. (2013), we mitigate this issue by employing *straight-through* estimator. More specifically, during backpropagation, the masking step is ignored and the gradient after the masking step flows "straight-through" to the step before the masking step. As a result, the gradient of loss $L$ with respect to $S_{i,j}$ and $W_{i,j}$ can be calculated as in Eq. (5) and (6), respectively.

$$\frac{\partial L}{\partial S_{i,j}} = \frac{\partial L}{\partial a_i} \frac{\partial a_i}{\partial S_{i,j}} = \frac{\partial L}{\partial a_i} W_{i,j} x_j \qquad (5)$$

$$\frac{\partial L}{\partial W_{i,j}} = \frac{\partial L}{\partial a_i} \frac{\partial a_i}{\partial W_{i,j}} = \frac{\partial L}{\partial a_i} M_{i,j} x_j \qquad (6)$$

From Eq. (6), we derive $\frac{\partial L}{\partial a_i} = \frac{\partial L}{\partial W_{i,j}} \frac{1}{M_{i,j}} \frac{1}{x_j}$. By omitting the binary mask term $M_{i,j}$ as in Sanh et al. (2020), we obtain $\frac{\partial L}{\partial a_i} = \frac{\partial L}{\partial W_{i,j}} \frac{1}{x_j}$. Inserting the obtained result of $\frac{\partial L}{\partial a_i}$ into Eq. (5) yields $\frac{\partial L}{\partial S_{i,j}}$ = $\frac{\partial L}{\partial W_{i,j}} \frac{1}{x_j} W_{i,j} x_j$. Therefore, the gradient of $L$ with respect to $S_{i,j}$ can be represented as $\frac{\partial L}{\partial S_{i,j}}$ = $\frac{\partial L}{\partial W_{i,j}} W_{i,j}$. The importance score $S_{i,j}$ after $T$ gradient updates can be represented as:

$$S_{i,j}^{(T)} = -\alpha_i \sum_{t<T} \left( \frac{\partial L}{\partial W_{i,j}} \right)^{(t)} W_{i,j}^{(t)}$$

where $T$ denotes the number of gradient updates, $\alpha_i$ is the learning rate during training process. In our method, a specific percentage of weights is pruned based on the distribution of importance score values, regardless of the absolute score values. The learning rate $\alpha_i$, which remains constant across all score parameters, does not impact the distribution and can be disregarded for simplicity without affecting the pruning outcome, as shown in Eq. (2).