# OpenReview forum: "Gradient-based Gradual Pruning for Language-Specific Multilingual Neural Machine Translation"
_EMNLP/2023/Conference — EMNLP 2023 Main_

### Official Review · Reviewer_29ZV · 2023-07-21

**Soundness:** 3

**Excitement:**

3: Ambivalent: It has merits (e.g., it reports state-of-the-art results, the idea is nice), but there are key weaknesses (e.g., it describes incremental work), and it can significantly benefit from another round of revision. However, I won't object to accepting it if my co-reviewers champion it.

**Missing References:**

- [1]: [Learning Language-Specific Layers for Multilingual Machine Translation](https://aclanthology.org/2023.acl-long.825) (Pires et al., ACL 2023)
- [2]: [Lifting the Curse of Multilinguality by Pre-training Modular Transformers](https://aclanthology.org/2022.naacl-main.255) (Pfeiffer et al., NAACL 2022)
- [3]: [Results of WMT22 Metrics Shared Task: Stop Using BLEU – Neural Metrics Are Better and More Robust](https://aclanthology.org/2022.wmt-1.2) (Freitag et al., WMT 2022)

**Paper Topic And Main Contributions:**

The presented paper proposes a gradient-based gradual pruning approach for multilingual machine translation that will extract language-pair specific subnetworks to boost translation quality. Additionally, different sub-network extraction techniques are compared and  the overlap in sub-networks is analyzed showing that sub-network overlap is correlated with language similarity.

**Questions For The Authors:**

**Question A**: The additional sub-network search procedure e.g. Phase 2 + 3 imposes computational overhead during training that could be used differently e.g. training the baselines for longer. I'm curious how that will affect the comparison?

**Question B**: Will code be released for the experimental setup? Specifically, Phase 2 and Phase 3 would be interesting to the community I believe

**Reasons To Accept:**

- Improves two weaknesses in Lin et al. (2021) i.e. one-shot + magnitude pruning and clearly differentiates the novelty of the presented work
- Very well written and easy to follow the main idea and experiments
- Interesting ablation on the pruning ratio which verifies that gradual gradient pruning is very effective

**Reasons To Reject:**

- **[large]**: Besides LaSS the presented method is not compared to any other method that has been proposed for MNMT. I'd appreciate a more thorough comparison to existing literature. Specifically, I'd recommend to compare against LSL-NAS + dense pre-training from [1], X-MOD from [2], or Adapter-based approaches, similar to what is done in [1]. All of these share similar efficiency advantages to what is presented here since they make use of sparsity.
- **[medium]**: While BLEU is still widely used, there are now better metrics to use for Machine Translation that correlate better with human judgment [3], specifically I'd recommend including chrF and COMET scores.
- **[medium]**: Since sub-layers are extracted for specific language pairs, it would be interesting to see how that affects zero-shot performance. I'm assuming it will be very detrimental to that since it is unclear which subnetwork to use so I'd consider this a limitation of this approach over the dense baseline.
- **[large]**: The proposed method requires one sub-network mask $\mathbf{M}\_{s\_i \rightarrow t\_i }$ per language-pair supported in the MNMT model. In the many-to-many (M2M) scenario, this will require $L * (L-1)$ masks (where $L$ is the number of languages supported in the model). Assuming we apply the proposed approach for all non-embedding parameters in the model, each mask will have the size $|\theta_0| - |\theta_\{embed\}|$ for a total of $L * (L-1) * (|\theta_0| - |\theta_\{embed\}|)$ entries that will require significant disk space to store during inference. With a high pruning ratio, we can resort to only storing the active weight indices but scaling this to hundreds of languages seems sub-optimal given the quadratic growth. Since the proposed approach is motivated for "resource-constrained scenarios" (l. 161), I fail to see how this is more resource efficient in terms of disk space than training sparse lower-dimensional language-specific modules (which will scale linearly in $L$ and still maintain a similar efficient inference costs and memory consumption). **As a result: I believe a more thorough analysis regarding disk requirements and inference speed is required across different methods.**

**Reproducibility:**

4: Could mostly reproduce the results, but there may be some variation because of sample variance or minor variations in their interpretation of the protocol or method.

**Reviewer Confidence:**

4: Quite sure. I tried to check the important points carefully. It's unlikely, though conceivable, that I missed something that should affect my ratings.

**Typos Grammar Style And Presentation Improvements:**

- Algorithm 1 l.3 missing space before $\theta_0$
- Please include all figures as vector graphics so they look good in print and scale properly e.g. Figure 1, 2, 3
- Some of the cited bibliography entries have published version, e.g. Han et al., 2015 was published at ICLR 2016. Please go through your cited sources and ensure to credit the published version.

---

> ### Author Rebuttal · Authors · 2023-08-29
>
> Dear Reviewer 29ZV,
>
> Thank you for your careful review of our paper. We greatly appreciate your feedback and address your concerns and questions below.
>
> >**1.  [large]: Besides LaSS the presented method is not compared to any other method that has been proposed for MNMT. I'd appreciate a more thorough comparison to existing literature. Specifically, I'd recommend to compare against LSL-NAS + dense pre-training from [1], X-MOD from [2], or Adapter-based approaches, similar to what is done in [1]. All of these share similar efficiency advantages to what is presented here since they make use of sparsity.**
> >
>
> Thank you for your constructive insights. The first paper you mentioned from ACL 2023 is intriguing. However, it became publicly available on ACL around July, whereas we submitted our paper in June. We will soon expand our related work section to mention it and also the second paper you pointed out.
>
> However, in this study, we aim to prevent the inclusion of any trainable parameters in the model and avoid inference speed slowdown. Thus, we chose the method LaSS, which shares a similar philosophy, for comparison. It is essential to highlight that the mask matrices in our method are obtained based on the specific pruning criterion and schedule. They are not learnable parameters.
>
> Designing language-specific parameters by adding extra parameters to the model could lead to two issues:  1) an unfair comparison when compared with techniques and models with fewer parameters. 2) a decrease in inference speed. More specifically, Adapter-based approaches commonly introduce learnable parameters for each language pair, thus increasing the parameter count for the entire model of all languages and the parameter count for individual language pairs, causing inference speed decrease. Designing distinct layers/modules for different languages in the two mentioned papers maintains a constant size for each pair and has no/negligible impact on inference speed. However, it still leads to the augmentation of the total model parameters.
>
>
>
>
> > **2. [medium]: While BLEU is still widely used, there are now better metrics to use for Machine Translation that correlate better with human judgment [3], specifically I'd recommend including chrF and COMET scores.**
> >
>
> Thanks for your feedback on the evaluation metric for Machine Translation. While COMET has gained recognition for its robust correlations with human judgments, recent research (Identifying Weaknesses in Machine Translation Metrics Through Minimum Bayes Risk Decoding: A Case Study for COMET) shows that it can be unreliable in identifying specific critical errors, such as deviations in entities and numbers. As a result, we have opted to utilize the (most) widely employed evaluation metric BLEU, combined with the win ratio, to evaluate the performance of different methods.
> Nonetheless, we greatly value your recommendation and fully agree that considering COMET holds value. We report the COMET scores and chrF scores on both the IWSLT dataset and WMT dataset below. The results in Table 1 and Table 2 below confirm the improvements of our approach over the baseline and MagnitudeOneshot (LaSS) approach reported in Tables 1 and 3 in the paper. We will upload these results in the final version of the paper.
>
> Table 1 IWSLT chrF and COMET performance:
> | | chrF |  COMET  |
> |------------------------ |------------|----------|
> | Baseline                                      | 51.2 | 0.773 |
> |MagnitudeOneshot (LaSS)             | 51.5 | 0.775 |
> | GradientGradual (Ours)            | **52.9** | **0.788** |
>
>
> Table 2  WMT chrF and COMET performance:
> | | chrF |  COMET  |
> |------------------------ |------------|----------|
> | Baseline                                      | 44.4 | 0.737 |
> |MagnitudeOneshot (LaSS)             | 45.4 | 0.747 |
> | GradientGradual (Ours)            | **45.9** | **0.755** |
>
>
>
> >**3. [medium]: Since sub-layers are extracted for specific language pairs, it would be interesting to see how that affects zero-shot performance. I'm assuming it will be very detrimental to that since it is unclear which subnetwork to use so I'd consider this a limitation of this approach over the dense baseline.**
> >
>
> Thanks for sharing your input on zero-shot. We argue that our method has the capability to surpass the performance of the dense baseline. It is demonstrated by Lin et al. (2021) that by combining the encoder mask of X -> En and the decoder mask of En -> Y for X -> Y translation direction,  LaSS obtains consistent gains over the multilingual baseline in all language pairs for zero-shot evaluation. Meanwhile, as you described in the review feedback, our approach improves two weaknesses in Lin et al. (2021), and the ablation study on the pruning ratio further verifies that our method is more effective. Therefore, our approach has a strong probability of outperforming the dense baseline. Due to the constraints of paper dimensions and the limited time for the author response, we decided to leave the exploration of our approach in this direction to future work.
>
>
> >**4. [large]: …With a high pruning ratio, we can resort to only storing the active weight indices but scaling this to hundreds of languages seems sub-optimal given the quadratic growth. Since the proposed approach is motivated for "resource-constrained scenarios" (l. 161), I fail to see how this is more resource efficient in terms of disk space than training sparse lower-dimensional language-specific modules (which will scale linearly in L(latic form) and still maintain a similar efficient inference costs and memory consumption). As a result: I believe a more thorough analysis regarding disk requirements and inference speed is required across different methods.**
> >
>
>
>
> Thanks for your feedback. We appreciate your input. We clarify the disk requirements and the inference speed with detailed evaluation results below.
>
> **Response to the definition of “resource-constrained”:** By "resource-constrained", we mainly refer to scenarios where researchers do not have enough data and computational resources to train the big model when a significant number of additional language-specific parameters are introduced. For instance, the design of separate decoders for each language pair following the conventional transformer structure, with an equal number of encoder and decoder layers, will potentially require more data and computational resources to successfully train all the parameters. We will improve the clarity of this specific sentence to avoid potential confusion.
>
>
> **Response to disk requirement:**
> Regarding disk requirements, we would like to clarify the following points.
> Our approach is not applied to any bias parameters and layer normalization layers. Assuming the size of each mask in our approach is |$\theta_m$|, |$\theta_m$|is supported to be less than (|$\theta_0$|-|$\theta_{embed}$|). Moreover, as you mentioned, we can exclusively store indices of active weights,  reducing the size to less than |$\theta_m$|/2. Additionally, compared to the extra parameters introduced to the model in other studies, typically stored in float32 format and requiring 4 bytes per element, each element in our approach requires only 1 byte.
> Further, in this work, we focus on English-centric multilingual translation and leave the exploration of M2M to future work. However, for the M2M scenario, our strategy will not be using $L*(L-1)$ masks for $L$ languages. Instead, we would utilize a single encoder mask for all the language pairs with the same source language and a single decoder mask for all pairs with the same target language, which equals requiring $L$ distinct full masks in terms of disk storage requirement (a full mask contains an encoder mask and a decoder mask). An alternative option is to search for encoder and decoder masks on a language-family basis, suggesting less than $L$ different full masks are required. Further investigation is needed for a comprehensive understanding, and we would like to extend our effort in this direction in future work, as mentioned in our limitation section. With the clarifications above, the impact of our approach on disk space is manageable, even in the M2M scenario.
>
>
>
>
>
> **Regarding inference speed, we provide clarification below, accompanied by detailed results, to demonstrate that our approach has a negligible to no impact on inference speed.**
>
> The subnetwork mask matrices are represented as binary matrices and are obtained during training and directly applied to Attention and Feedforward sub-layers during inference. Compared to the compute-intensive multiplication between dense weight matrices (typically in float32 or float16), multiplying a binary mask matrix with a weight matrix on some sub-layers introduces a  minimal overhead and, therefore, has a negligible impact on inference speed.
> In addition, zero elements in the masked weight matrix could result in faster inference speed due to the possibility of avoiding unnecessary arithmetic operations; for example, $x+0$ is always $x$, and $x*0$ is always 0. Furthermore, our approach provides the potential for making the most of model sparsity and further boosting inference speed through advanced techniques, such as Sparse Multiplication Optimization.
>
> Following your suggestion, we present the inference speed results of the MNMT baseline and our method without applying additional techniques such as Sparse Multiplication for a fair comparison.
>
> We report tokens/second on the de-en test set with Transformer-small(details shown in Appendix B Model details section). Following the inference speed evaluation setup in Pires et al. (2023), the batch size is always set to 1 and the result is averaged over 5 runs. Speed GPU is measured using a single NVIDIA A100 GPU and   Speed CPU is measured using a single-threaded Intel(R) Xeon(R) Gold 6330 CPU @ 2.00GHz.
>
> While the best performance of our approach is achieved with pruning_ratio set to 0.6, as shown in Figure 1 in the paper, we provide the inference speed results with the pruning ratio ranging from 0.1 to 0.9. We hope this could offer a more comprehensive analysis of inference speed. Although Speed GPU  and CPU are different in absolute value, they show a similar pattern across different pruning ratios, and we take the Speed GPU as an example to analyze.
> The results indicate an improvement in the inference speed of our approach when compared with the multilingual baseline within the pruning_ratio range of 0.6 to 0.9, and the highest inference speed is obtained when pruning_ratio is set to 0.9. We attribute this speed improvement to the avoidance of unnecessary operations with element 0, as explained above. Additionally, the inference speed is slightly slower than the MNMT baseline, with pruning_ratio smaller than 0.6, but not statistically significantly different.
>
> | Model                              |Speed GPU (tokens/s)   |Speed CPU (tokens/s) |
> |------------------------ |------------|----------|
> | Multilingual Baseline               |237.56 ± 2.70   |59.98 ± 2.73 |
> | GradientGradual( pruning_ratio=0.1) |235.23 ± 3.65   |59.57 ± 0.77 |
> | GraientGradua( pruning_ratio=0.2)   |233.93 ± 3.28   |58.81 ± 4.27 |
> | GraientGradua( pruning_ratio=0.3)   |235.71 ± 1.92   |58.26 ± 3.34 |
> | GraientGradua( pruning_ratio=0.4)   |236.35 ± 2.71   |58.04 ± 2.93 |
> | GraientGradua( pruning_ratio=0.5)   |235.94 ± 2.08   |59.37 ± 1.73 |
> | GraientGradua( pruning_ratio=0.6)   |238.76 ± 2.49   |58.94 ± 3.28 |
> | GraientGradua( pruning_ratio=0.7)   |240.10 ± 2.62   |60.17 ± 2.70 |
> | GraientGradua( pruning_ratio=0.8)   |242.55 ± 2.47   |60.58 ± 2.37 |
> | GraientGradua( pruning_ratio=0.9)   |243.23 ± 2.89   |61.56 ± 3.99 |
>
>
>
> **Regarding your questions**
> > **Question A: The additional sub-network search procedure e.g. Phase 2 + 3 imposes computational overhead during training that could be used differently e.g. training the baselines for longer. I'm curious how that will affect the comparison?**
> >
>
> The multilingual baseline model serves as the initialization for the final model. The baseline and the final model can indeed exhibit different absolute performances depending on the duration of baseline training. With the same baseline(initialization), the quality of the final model relies on the capability of extracting high-quality sub-networks for language pairs. The robustness results in section 5.3 and the similarity scores in section 5.5 in the paper confirmed the effectiveness and robustness of our approach compared to MagnitudeOneshot. Therefore, we believe that as long as the baseline model is the same, the relative quality of the final models remains the same regardless of the duration of the baseline training (GradientGradual > MagnitudeOneshot > Baseline).
>
>
> Specifically, in our experiment, we configured the hyper-parameter “max_update” to a sufficiently large value to avoid underfitting and employed the early stop technique by configuring the hyper-parameter “patience” to 30 to prevent overfitting. We aimed to train an optimal baseline, serving as the foundation for the final model for both GradientGradual and MagnitudeOneshot. In this way, the final model can potentially achieve peak performance. This configuration simulates the most likely scenario used to achieve the best MNMT model in real-world applications.
>
> >**Question B: Will code be released for the experimental setup? Specifically, Phase 2 and Phase 3 would be interesting to the community I believe**
> >
>
> We appreciate your interest, and we fully recognize the value of sharing code with the community. We are actively working to explore the possibility of releasing both the code and experimental setups.
>
>
> **Regarding your feedback on Grammar Style and Presentation improvement**
>
> > **Algorithm 1 l.3 missing space before $\theta_0$ and other feedback.**
> >
>
> Thanks for your careful review. We will correct each Grammar issue you pointed out and improve our presentation soon in our paper. We will also extend our related work section to mention the missing references you pointed out.

---

### Official Review · Reviewer_KqQ9 · 2023-08-02

**Soundness:** 4

**Excitement:**

3: Ambivalent: It has merits (e.g., it reports state-of-the-art results, the idea is nice), but there are key weaknesses (e.g., it describes incremental work), and it can significantly benefit from another round of revision. However, I won't object to accepting it if my co-reviewers champion it.

**Paper Topic And Main Contributions:**

Due to parameter interference when parameters are fully shared across all language pairs, the traditional multilingual neural machine translation often suffers from performance degradation in high-resource languages compared to bilingual counterparts. To tackle this issue, they propose a gradient-based gradual pruning technique for MNMT. Compared with the previous similar work (Lin et al,2021), they use gradient-based scores instead of magnitude scores as  pruning criterion and use gradual pruning instead of one-shot pruning as pruning schedule. Experiments on IWSLT and WMT datasets show the effectiveness of this method.

**Questions For The Authors:**

Could you provide the detailed results for every language pair on IWLST and WMT? Table 1 provides only the average en<->x BLEU score. Table 3  doesn't even give the BLEU score for every language pair.

**Reasons To Accept:**

This paper is well-written and easy to follow.
This paper provided the analysis to show the impact of four pruning methods in the evaluation of model performance.

**Reasons To Reject:**

The novelty is limited. This work is similar with the previous work (Lin et al,2021).
Please provide the detailed results.


**Reproducibility:**

4: Could mostly reproduce the results, but there may be some variation because of sample variance or minor variations in their interpretation of the protocol or method.

**Reviewer Confidence:**

3: Pretty sure, but there's a chance I missed something. Although I have a good feel for this area in general, I did not carefully check the paper's details, e.g., the math, experimental design, or novelty.

---

> ### Author Rebuttal · Authors · 2023-08-29
>
> Dear Reviewer KqQ9,
>
> We highly appreciate your feedback on our paper, and we address your concerns and questions below.
>
> >1. The novelty is limited. This work is similar with the previous work (Lin et al,2021). Please provide the detailed results.
> >
>
> **Regarding novelty:** Thanks for your feedback and your insight is highly appreciated.
> Regarding novelty, as noticed by reviewer 29ZV, our approach is unique in exploring the effectiveness of various pruning criteria and schedules within the context of multilingual neural machine translation. This aspect has received limited attention in LaSS (Lin et al. 2021) and remains entirely unexplored in other prior work. We kindly invite you to check our detailed explanation of a similar question posed by the reviewer ro9B.
>
> **Regarding the detailed results:** Thank you for your careful review and valuable feedback. Considering the relatively extensive number of languages involved in the wmt dataset, we have adopted a strategy similar to several prior research studies by presenting average scores for low-, mid-, and high-resource groups. This way, it offers an overview of performance in different data situations. However, your feedback is greatly appreciated, and we accordingly provide detailed scores below. We first report the averaged performance results of En->X and X->En for low- medium- and high-resourced languages  (Tables 1, 4, and 7) and then present the performance splitting En->X and X->En in all the data conditions (Tables 2, 3, 5, 6, 8, and 9). The results confirm the findings we report in the paper, where the gain over the baseline and LaSS improve by increasing the data conditions. Interestingly, we also observed that the major gains our approach and LaSS have over the baseline are in the En->X directions. This suggests that the multilingual decoder benefits more from the application of subnetworks. This aspect has not been noticed in previous papers.
>
>
> (1) Low-resource languages: Gu (7.5 K), Kk (102 K), Tr (198 K), Ro (596K) and Ta (605 K)
>
> Table 1: The average En <-> X performance for low-resource languages:
> |Lang                           | Gu |  Kk  | Tr | Ro |  Ta  | average|
> |--------------------------------|------------  |----------|----------|----------|----------|----------|
> | Baseline                          | 0.75| 6.00 | 19.55|28.85|14.90|14.01|
> |MagnitudeOneshot            | 0.25| 0.40 | 0.65  |0.50  |1.70| 0.70|
> |GradientGradual (Ours) | -0.10| -0.55 | 0.75  |1.35  |3.10| **0.91**|
>
>
> Table 2: En->X performance for low-resourced languages:
> |Lang                           | Gu |  Kk  | Tr | Ro |  Ta  | average|
> |--------------------------------|------------  |----------|----------|----------|----------|----------|
> |Baseline|1.20|2.50|17.80|31.10|14.10|13.34|
> |MagnitudeOneshot|0.10|1.30|1.70|0.80|3.10|1.40|
> |GradientGradual (Ours)(ours)|-0.20|1.40|1.70|1.90|6.00|**2.16**|
>
> Table 3: X->En performance for low-resourced languages:
> | Lang                          | Gu |  Kk  | Tr | Ro |  Ta  | average|
> |--------------------------------|------------  |----------|----------|----------|----------|----------|
> |Baseline|0.30|9.50|21.30|26.60|15.70|14.68|
> |MagnitudeOneshot|0.40|-0.50|-0.40|0.20|0.30|**0.00**|
> |GradientGradual (Ours)|0.00|-2.50|-0.20|0.80|0.20|-0.34|
>
> (2) Medium-resource languages: Ps(1M), Fi(2M), Lv(2.1M), Et(2.1M), Ru(2.3M), Lt(3.4M) and De(4.4M)
>
> Table 4: The average En <-> performance for medium-resource languages:
> | lang                     |Ps|Fi|Lv|Et|Ru|Lt|De|average|
> |--------------------------------|------------  |----------|----------|----------|----------|----------|-------|-------|
> |Baseline|6.95|18.40|16.45|18.75|25.45|12.85|23.90|17.54|
> |MagnitudeOneshot|0.20|1.10|0.75|0.90|1.45|0.20|1.35|0.85|
> |GradientGradual (Ours)|1.85|1.50|1.30|1.75|2.30|0.15|2.10|**1.56**|
>
> Table 5: The En->X  performance for medium-resource languages:
> | lang                     |Ps|Fi|Lv|Et|Ru|Lt|De|average|
> |--------------------------------|------------  |----------|----------|----------|----------|----------|-------|-------|
> | Baseline|4.10|14.70|14.10|15.10|22.20|8.60|19.90|14.10|
> | MagnitudeOneshot|0.40|1.80|1.10|1.30|2.40|0.10|2.00|1.30|
> | GradientGradual (Ours)|2.60|2.30|2.00|2.30|3.50|0.00|2.90|**2.23**|
>
>
> Table 6: The X->En performance for medium-resource languages:
> | lang                     |Ps|Fi|Lv|Et|Ru|Lt|De|average|
> |--------------------------------|------------  |----------|----------|----------|----------|----------|-------|-------|
> |Baseline|9.80|22.10|18.80|22.40|28.70|17.10|27.90|20.97|
> |MagnitudeOneshot|0.00|0.40|0.40|0.50|0.50|0.30|0.70|0.40|
> |GradientGradual (Ours)|1.10|0.70|0.60|1.20|1.10|0.30|1.30|**0.90**|
>
> (3) High-resource languages: Cs(1M), Pl(2M), Es(2.1M), Ja(2.1M), Zh(2.3M) and Fr(3.4M)
>
> Table 7: The average En <-> X performance for high-resource languages:
> | lang                     |Cs|Pl|Es|Ja|Zh|Fr|average|
> |--------------------------------|------------  |----------|----------|----------|----------|----------|-------|
> |Baseline|24.05|20.35|30.40|17.90|19.10|33.15|24.16|
> |MagnitudeOneshot|0.70|1.20|0.75|1.00|1.20|1.10|0.99|
> |GradientGradual (Ours)|1.25|1.15|1.55|1.75|2.75|1.55|**1.67**|
>
> Table 8: The En->X  performance for high-resource languages:
> | lang                     |Cs|Pl|Es|Ja|Zh|Fr|average|
> |--------------------------------|------------  |----------|----------|----------|----------|----------|-------|
> |Baseline|19.40|15.90|29.40|20.00|22.70|32.60|23.33|
> |MagnitudeOneshot|1.90|1.60|1.10|1.20|0.70|1.80|1.38|
> |GradientGradual (Ours)|2.70|1.60|2.00|2.30|3.00|2.50|2.35|
>
> Table 9: The X->En performance for high-resource languages:
> | lang                     |Cs|Pl|Es|Ja|Zh|Fr|average|
> |--------------------------------|------------  |----------|----------|----------|----------|----------|-------|
> |Baseline|28.70|24.80|31.40|15.80|15.50|33.70|24.98|
> |MagnitudeOneshot|-0.50|0.80|0.40|0.80|1.70|0.40|0.60|
> |GradientGradual (Ours)|-0.20|0.70|1.10|1.20|2.50|0.60|**0.98**|
>
>
>
>
> >2. Could you provide the detailed results for every language pair on IWLST and WMT? Table 1 provides only the average en<->x BLEU score. Table 3 doesn't even give the BLEU score for every language pair.
> >
> Thank you for your feedback. We provide the detailed scores in the above section and will include them in the appendix section of our paper soon.

---

### Official Review · Reviewer_ro9B · 2023-08-05

**Soundness:** 4

**Excitement:**

4: Strong: This paper deepens the understanding of some phenomenon or lowers the barriers to an existing research direction.

**Paper Topic And Main Contributions:**

The paper tries to improve multilingual NMT models by addressing the parameter interference problem using a  gradient-based gradual pruning approach.

Main contributions:
- significant improvements in BLEU;
- ablation study on pruning criteria and schedules;
- ablation study on  the contribution of various sub-layers;

**Questions For The Authors:**

- could the authors add another baseline which uses bilingual models? i.e., training a bilingual model for each language pair. This baseline can be used to answer how much the proposed approach can fix the parameter interference problem for high-resource language pairs, e.g., ideally the proposed approach should outperform the bilingual baselines on all the language pairs.

**Reasons To Accept:**

- the improvements are promising;
- the paper is well written;

**Reasons To Reject:**

- the proposed approach has multiple steps which may be hard to be used or followed by future work;
- the general direction is not novel, i.e., extract sub-layers for bilingual language pairs;

**Reproducibility:**

3: Could reproduce the results with some difficulty. The settings of parameters are underspecified or subjectively determined; the training/evaluation data are not widely available.

**Reviewer Confidence:**

3: Pretty sure, but there's a chance I missed something. Although I have a good feel for this area in general, I did not carefully check the paper's details, e.g., the math, experimental design, or novelty.

---

> ### Author Rebuttal · Authors · 2023-08-29
>
> Dear Reviewer ro9B,
>
> We are extremely grateful for your valuable feedback on our paper, and we will address each of your concerns and questions in detail below.
>
> >1. the proposed approach has multiple steps which may be hard to be used or followed by future work;
> >
> Thank you for your input on the training processing. In this work, we aim to explore the most effective approach for mitigating the parameter interference issue in the multilingual NMT model and improving the model performance. We acknowledge that the three-phase training process might introduce
> additional complexity to the training pipeline compared to the standard MNMT training. Moving forward, we plan to explore the possibility of making the training process end-to-end in our future work.
>
>
> >2. the general direction is not novel, i.e., extract sub-layers for bilingual language pairs;
> >
>  Thanks for sharing your perspective on the novelty of our work. We understand that extracting sub-networks for bilingual language pairs is not a new concept. However, the critical challenge of pinpointing essential parameters for each language pair remains relatively underexplored.
>
> In addition, although pruning is a commonly employed technique in machine learning, there is a significant lack, or complete absence, of exploration regarding effective pruning in multilingual neural machine translation. The most widely used pruning criterion and schedule can be ineffective for this specific task. For instance, we observed a high overlap of retained weights across different language pairs when utilizing magnitude pruning for sub-network extraction. Consequently, the distinct and conflicting requirements on the extensively shared parameters of various language pairs persist, thus limiting the effectiveness of parameter interference mitigation.
>
> By researching pruning criteria and schedules within multilingual neural machine translation, we aspire to shed light on exploring the most effective approach in mitigating parameter interference across languages while not introducing additional parameters to the model for future MNMT research work.
>
> >3. could the authors add another baseline which uses bilingual models? i.e., training a bilingual model for each language pair. This baseline can be used to answer how much the proposed approach can fix the parameter interference problem for high-resource language pairs, e.g., ideally the proposed approach should outperform the bilingual baselines on all the language pairs.
> >
>
> Thank you for your suggestion. Following your advice, we conducted experiments using bilingual models, and we hope this can offer insights into the performance comparison of utilizing a bilingual model against our method with the multilingual model.
> Due to the limited time for the author response, we selected one language from each of the low-, medium-, and high-resource groups to conduct bilingual experiments. The sizes of our chosen languages fall in the middle range of each respective group’s size spectrum. We conducted bilingual experiments in En->X and X->En translation directions and present the average En<->X BLEU scores for Turkish (Tr), Estonian(Et), and Chinese(Zh) below.
>
> Table 1: The average BLEU scores of En <-> Tr (198K), En <-> Et (2.1 M), and  En <-> Zh (17.8M).
> | Lang | Tr |  Et  | Zh |
> |------------------------ |------------|----------|----------|
> | Bilingual                          | 11.9 | 18.55 | **21.9** |
> | MNMT baseline              | 19.55 | 18.75 |19.1 |
> | LaSS                                | 20.2 | 19.65 | 20.3|
> | GradientGradual (Ours) | **20.3** | **20.5** | 21.85 |
>
> **MNMT baseline versus Bilingual:** The results show that the MNMT baseline significantly outperforms the bilingual model in the low-resource language by 7.65 (19.55 vs. 11.9) BLEU scores, achieves similar performance to the bilingual model (18.75 vs. 18.55) in the medium-resource language and experiences a considerable decrease of 2.8 (19.1 vs. 21.9) BLEU scores in the high-resource language. This observation aligns with our experience that while low-resource languages often benefit from knowledge transfer in MNMT, high-resource languages can suffer from performance degradation due to the lack of language-specific parameters and the issue of parameter interference resulting from full parameter sharing in MNMT.
>
> **LaSS and GradientGradual (Ours)  versus  Bilingual:** In the low-resource language, both LaSS and our approach significantly outperform the bilingual model, with 8.3 (20.2 vs. 11.9) BLEU scores and 8.4 (20.3 vs. 11.9) BLEU scores, separately. In the medium-resource language,  while LaSS exhibits a performance gain of 1.1 (19.65 vs. 18.55) BLEU scores,  our approach achieves a larger improvement of 1.95 (20.5 vs. 18.55) BLEU scores. Notably, LaSS underperforms the baseline in the high-resource language by a notable margin of 1.6 (20.3 vs. 21.9)  BLEU scores. In contrast, our approach achieves similar performance to the bilingual model (21.85 vs. 21.9) in the high-resource language.
>
> **Summarization:** Our method shows a larger improvement over the Bilingual models than the MNMT baseline and LaSS in low- and medium-resource languages. Furthermore, while the MNMT baseline and LaSS experience performance degradation compared to the Bilingual models in the high-resource language, our approach demonstrates almost the same performance (21.85 vs. 21.9), alleviating the typical problem of performance degradation for high-resource languages in the MNMT model.

---

### Meta-Review · Area_Chair_os3x · 2023-09-23

**Recommendation:** 4

**Metareview:**

The overall excitement of the reviewers was ambivalent, but the active author response discussion period showed a lot more positive sentiment from the reviewers on the quality of the work especially with the addition of the extra experimental results provided by the authors and the extra context and thoughtful reasoning provided to answer the questions raised by the reviewers.

While the overall assessment is higher than the numerical scores assigned by the reviewers, this is mainly due to the extra information that was provided during the author response, so it is very important that all of the extra experimental results and clarifications are included as part of the paper by the authors. The additional context provided by the information in the author response is a big reason for the final evaluation of this submission in this meta-review.

---

### Decision · Program_Chairs · 2023-10-07

**Decision:**

Accept-Main

**Comment:**

The overall excitement of the reviewers was ambivalent, but the active author response discussion period showed a lot more positive sentiment from the reviewers on the quality of the work especially with the addition of the extra experimental results provided by the authors and the extra context and thoughtful reasoning provided to answer the questions raised by the reviewers.

While the overall assessment is higher than the numerical scores assigned by the reviewers, this is mainly due to the extra information that was provided during the author response, so it is very important that all of the extra experimental results and clarifications are included as part of the paper by the authors. The additional context provided by the information in the author response is a big reason for the final evaluation of this submission in this meta-review.